# Limited evidence for executive function load impairing selective copying in a win-stay lose-shift task

Juliet Dunstone[1,2]*, Mark Atkinson[1,2], Catherine Grainger[1], Elizabeth Renner[1,2], Christine A. Caldwell[1,2]

1 Psychology Division, University of Stirling, Stirling, United Kingdom, 2 RATCHETCOG Research Group, University of Stirling, Stirling, United Kingdom

* julietdunstone@gmail.com

**Data Availability Statement:** Relevant data and scripts to replicate analysis are within the manuscript and its Supporting information files. Scripts to replicate the study tasks are available in

## Abstract

The use of 'explicitly metacognitive' learning strategies has been proposed as an explanation for uniquely human capacities for cumulative culture. Such strategies are proposed to rely on explicit, system-2 cognitive processes, to enable advantageous selective copying. To investigate the plausibility of this theory, we investigated participants' ability to make flexible learning decisions, and their metacognitive monitoring efficiency, under executive function (EF) load. Adult participants completed a simple win-stay lose-shift (WSLS) paradigm task, intended to model a situation where presented information can be used to inform response choice, by copying rewarded responses and avoiding those that are unrewarded. This was completed alongside a concurrent switching task. Participants were split into three conditions: those that needed to use a selective copying, WSLS strategy, those that should always copy observed information, and those that should always do the opposite (Expt 1). Participants also completed a metacognitive monitoring task alongside the concurrent switching task (Expt 2). Conditions demanding selective strategies were more challenging than those requiring the use of one rule consistently. In addition, consistently copying was less challenging than consistently avoiding observed stimuli. Differences between selectively copying and always copying were hypothesised to stem from working memory requirements rather than the concurrent EF load. No impact of EF load was found on participants' metacognitive monitoring ability. These results suggest that copying decisions are underpinned by the use of executive functions even at a very basic level, and that selective copying strategies are more challenging than a combination of their component parts. We found minimal evidence that selective copying strategies relied on executive functions any more than consistent copying or deviation. However, task experience effects suggested that ceiling effects could have been masking differences between conditions which might be apparent in other contexts, such as when observed information must be retained in memory.

an OSF repository which is linked within the Supporting information or can be accessed directly here: https://osf.io/bfg3x/.

**Funding:** Recipient: CAC 648841 RATCHETCOG ERC-2014-CoG European Research Council https://erc.europa.eu/ The funders had no role in study design, data collection and analysis, decision to publish, or preparation of the manuscript.

**Competing interests:** The authors have declared that no competing interests exist.

## Introduction

The Explicitly Metacognitive Cumulative Culture hypothesis (EMCC; [1]) theorises that cumulative cultural evolution may rely on human-unique cognitive capacities for system-2 processes or explicit metacognition. These cognitive capacities are hypothesised to be a requirement for cumulative culture as they allow for efficient and adaptive use of selective copying strategies [2]. To investigate this theory, participants' ability to make efficient, flexible learning decisions while under additional cognitive load was examined.

Cumulative cultural evolution (CCE) is the process by which cultural traits (including behaviours, artefacts and tools) change over generations to become more effective and beneficial to their users [3]. This is generally thought to be unique to humans [4], and can lead to cultural traits evolving over many generations which could not have been invented by a single individual within their lifetime.

Although it is not possible to capture all of cumulative cultural evolution in a single cognitive capacity, and few (if any) authors would claim to have found a single trait which was responsible for uniquely human cumulative culture [5] many theories suggest possible abilities that could help facilitate this capacity in humans.

One current theory for why humans possess this unique capacity asserts that we have the ability to strategically apply social learning in adaptive contexts to ensure optimal performance; retention of beneficial information and innovation of new behaviours (to replace unhelpful or maladaptive elements of traits). The theory also proposes that this is possible because humans are explicitly aware of their learning strategies, and can strategically apply them in the most appropriate contexts. These learning strategies have been dubbed *explicitly metacognitive* social learning strategies (SLSs) by Heyes [2, 6] as they require thinking about mental states (knowledge of one's own knowledge, and sometimes also that of others). Explicitly metacognitive SLSs have been hypothesised to facilitate cumulative culture by allowing learners to direct social learning towards the most appropriate models [6], communicating metacognitive perspectives to in-group members [7] and making more effective use of information available in the environment [1].

Many social learning strategies, also referred to as learning biases or learning heuristics, have been identified (e.g. [8, 9]). However, these tend to refer to processes that are automatic, and often shared by non-human animals (henceforth animals). These SLS are referred to as 'planetary' by Heyes [2] as they can be described by observers, but generally do not exist in the mind of the animal or human carrying them out (a bit like laws of planetary motion). What distinguishes explicitly metacognitive SLS is that they are 'cook-like' [2] in that the rules to be followed are in the mind of the actor rather than (or as well as) the mind of the observer, similar to a cook following a recipe. Cook-like SLSs can be explicitly or verbally described by the agent using them, although it should be noted that the capacity to verbalise a strategy is not in itself the distinguishing feature between the metacognitive strategies, as this would exclude all non-humans by default. For a review of the non-human metacognition literature see [1].

It is the goal of this research to investigate these hypotheses by experimentally testing whether the capacity for explicitly metacognitive social learning strategies could indeed play an instrumental role in human cumulative culture. This will be assessed by attempting to restrict participants' access to system-2 cognitive resources (see section Dual-systems and dual-tasks) while they are required to make learning decisions that we assume to be necessary to generate cumulative culture. These learning decisions are argued to make CCE possible by improving upon actions already performed or observed.

## Testing capacities for cumulative culture

Multiple definitions of CCE have been given in the literature. Mesoudi and Thornton [5] have evaluated these definitions and identify four core elements that must be present for a particular case to be classified as CCE:

1. innovation or change to an established or observed behaviour

2. social learning of that behaviour by conspecifics

3. objective improvement of fitness (whether this is true biological fitness or the notion of 'cultural fitness') due to the socially learned behaviour change or innovation

4. repeated social learning of the new behaviour so that it outlives the original generation in which it appeared.

These criteria indicate that one necessary (but not sufficient) element of CCE is the capacity to retain the beneficial elements of a behaviour, but ignore or change the non-beneficial elements. At each stage of learning this would require a basic decision to either copy or repeat an action, or to perform a different action which may result in a more beneficial outcome. While in naturalistic settings this may provide a learner with near infinite possibilities of learning choices, this learning decision is essentially a win-stay, lose-shift (WSLS) paradigm taking place: observe successful behaviour (win) and do the same (stay), or observe unsuccessful behaviour (lose) and do something different (shift). This paradigm can therefore be experimentally modelled with a WSLS task.

Capturing individual learner behaviour in this way contrasts with experimental tasks which aim to capture CCE as a population-level process, over multiple learner generations. Methods such as the one used here can however be extended in ways that permit assessment of the potential for cumulative cultural evolution in contexts where experimentally examining generational turnover may be problematic [10].

The validity of the assumption that CCE relies upon the ability to adaptively switch between copying (staying) and exploring (shifting) can be tested by comparing participant performance in mixed blocks of trials where participants must flexibly choose between copying and exploring, with fixed blocks of trials in which participants only ever explore or only ever copy. The expected outcome of such comparisons would be that single trial-type conditions (always explore or always copy) would be rapidly automatized due to the same action being repeated every time, whereas the flexible trial conditions would not to the same extent. This would be taken to reflect reliance on automatic compared to explicitly metacognitive SLSs, respectively (due to highly practised, automatised actions being processed in a system-1 manner [11]).

## Dual-systems and dual-tasks

A key aspect of the hypothesis under investigation is that the metacognitive social learning strategies are explicitly applied, and used reflectively, with conscious access by the learner [2, 6]. It is this explicit use that likely differentiates explicitly metacognitive social learning strategies from the social learning strategies, learning biases or heuristics employed by animals such as those outlined in [9].

The concepts of implicit and explicit cognitive processes are often discussed without much consideration for specific theoretical definition [12]. To avoid this terminological black-boxing, this discussion will use dual processing theories (as summarised in [11]) to draw the distinction between implicit and explicit processes, with implicit processes being generally confined to system-1, or type-1, processing and explicit processes belonging to system-2 or type-2. The explicit nature of the processes means they rely on the use of executive functions–core cognitive

processes that are theorised to be key to system-2 processing. Metacognitive monitoring and control processes have also been shown to have strong correlates with executive function capacities in adults and children (see an extensive review by Roebers [13] for details).

This explicitly metacognitive hypothesis for the evolution of cumulative culture (EMCC) can therefore be tested with the use of dual tasks; secondary tasks completed concurrently with a main experimental task that put additional load onto certain executive functions. Due to the expectation of a processing bottleneck when multiple tasks are competing for the same executive function resources [14], if a concurrent executive function task impacts the ability to effectively apply efficient WSLS strategies, then executive function resources are implicated in the use of those strategies. Executive function dual tasks have been used, for example, to assess metacognitive involvement in an opt-out task [15], executive function requirements of Theory of Mind reasoning [16] and the implicit or explicit nature of certain processes such as perspective taking [17].

This study will therefore use executive function dual tasks to attempt to restrict access to metacognitive reasoning while participants complete a WSLS decision making task. Experiment 1 explores the impact that a dual-task has on participants' ability to apply a flexible WSLS strategy, using executive functions as a proxy for metacognition. Experiment 2 explores the impact of dual-tasks further, and also directly tests for the effect of dual-task interference on explicit metacognitive judgements. It should be noted that this task design does not implicate metacognition in the same way that a more complex task may have done. This decision was made intentionally in order to retain an easily manageable task that could capture a phenomenon which, despite not being sufficient for cumulative culture, we believe is almost certainly necessary.

The results of these studies are not intended to demonstrate the presence, or lack of, direct cumulative improvement over many generations. However, they are intended to model whether specific cognitive capacities allow or prevent learning decisions to be made that objectively improve upon observed behaviour.

**Pilot testing to establish a dual task paradigm.** Executive functions (EF) can be split into 3 main areas: storing, retrieving and updating information from working memory; inhibiting automatic responses to stimuli; and task switching [18]. Although correlated, these are distinct cognitive functions that affect different aspects of behaviour, as shown by confirmatory factor analyses [18, 19]. When designing dual-task methods to impede executive functions it would therefore be unwise to simply pick a secondary task without evaluating how the different EFs might interact with the main task.

Before a more substantial sample was recruited, multiple pilot tasks were tested on a smaller number of participants. The purpose of doing this was to compare the effect of multiple different executive function secondary tasks on a binary choice task (see below), which required the use of a WSLS strategy in order to perform correctly. This adopts a methodology similar to that used by Bull, Phillips and Conway [16], who tested multiple different executive functions (inhibition, switching and updating) with two theory of mind tasks, as well as a range of controls, to systematically assess the impact of executive function on theory of mind processing. Based on the outcome of this pilot, the subsequent studies presented in this paper employ a *task-switching* secondary task. Full details of the pilot study are given in the S1 and S2 Figs, S1–S3 Files and S1–S14 Data.

## Experiment 1

Dual task interference may be observable due to a reduced capacity to apply sustained attention to a rule that is currently required and change the response type accordingly, i.e., the

reduced ability to re-evaluate on each trial whether the presented information should be copied or not. This would predict significant interference from a secondary task when a selective strategy was required, but not when the same response type was required for every trial. System-2 processing is also implicated in the use of a selective response strategy, analogous to a basic selective SLS.

Experiment 1 therefore compared participant performance in a basic selection task when a selective strategy was required, compared to when it was not under conditions in which participants performed a secondary task that was intended to place a high or low load on executive resources. Mixed blocks of testing, where participants had to apply a selective response strategy (on some trials copy the provided information and on some trials do the opposite) were compared with blocks requiring single response-types (always copy or never copy). Participants in the always-copy and never-copy groups were shown information trials which were always successful or always unsuccessful, respectively, and therefore required responses of repetitive, reinforceable behaviours that do not require a system-2 SLS to be employed. Employing a blanket always-copy or never-copy could be considered strategies if they were applied to variable input. However, in the context of this experiment they are not, as they are not being used conditionally: In the WS (always copy—always apply a 'win-stay') condition participants only observe wins, and in the LS (never copy–always apply a 'lose-shift') condition participants only observe lose trials so the conditions themselves aren't strategic. Both always-copy and never-copy controls were included to differentiate between the difficulty increases of switching compared to copying, and flexibly copying compared to either copying or switching.

As only the mixed blocks required participants to continuously update and switch between response strategies, it was hypothesised that in this condition participants' performance would be negatively impacted by the secondary task, whereas interference would be minimal when participants were just using a repeated application of a single rule. It should be noted that this task, particularly in the single-response blocks, was designed to be simple for participants to do. This was partly to ensure that any effects found are not caused by confounding effects of main task difficulty or other cognitive factors that have not been strictly controlled for. The two control conditions were intended to capture a level of decision making where executive functions are *not* expected to be implicated. Pilot testing (see S1 and S2 Figs, S1–S3 Files and S1–S14 Data) demonstrated that dual-task interference did have a significant negative effect on responses in mixed blocks.

## Methods

**Participants.** Participants were recruited at the University of Stirling and took part in exchange for research participation tokens which were required for course completion. Twenty-four participants received cash remuneration instead and were paid at a rate of £5/ hour rounded up to the nearest pound. 166 participants took part (57 male, mean age: 21.2, age range: 16–51). Of these, 45 participants completed the training phase of the experiment but did not score above the inclusion threshold of 75% accuracy (see *procedure)* for the full experimental phase and so left the experiment after they had completed training. A further one participant was excluded because during debrief they informed the experimenter that they had not understood the task and had not been completing it correctly. In total 120 participants (47 male, mean age: 21.1, range: 16–49) completed the full study and were included in the dataset reported in the analyses below. All participants had normal or corrected to normal vision and hearing. Participants all gave written consent to take part and were aware that they were free to withdraw from the study at any time. Ethical approval for the study was given by the University of Stirling General University Ethics Panel (reference GUEP 111A). Participants

aged between 16–18 years old were treated as adults and gave their own consent, in line with British Psychology Society guidelines.

**Apparatus.**   Participants were tested using a desktop computer running Windows 8 with a standard mouse, a Black Box Toolkit 4 button response pad button box and Sony MDR-Pro over-ear headphones.

**Task design.**   All tasks were written and run in PsychoPy version 1.84.2 [20]. All participants completed the same main task, a binary choice task, in conjunction with a *task-switching* secondary task as well as a control secondary task. The dual task condition was a within-subjects variable, with each participant completing both the *switching* task and the control task, counterbalanced for order. All tasks are described below.

*Binary choice task.* This was a simple two alternative forced-choice (2AFC) binary choice task, intended to assess how quickly participants could use information they were presented with to make decisions. Participants were presented with an information trial which showed two shape stimuli on screen. After a brief pause the task revealed the value (successful–a fish was shown -or unsuccessful–a shark was shown) of one of the two stimuli. Participants were then required to make one selection from the two stimuli presented again (see Fig 1, and the

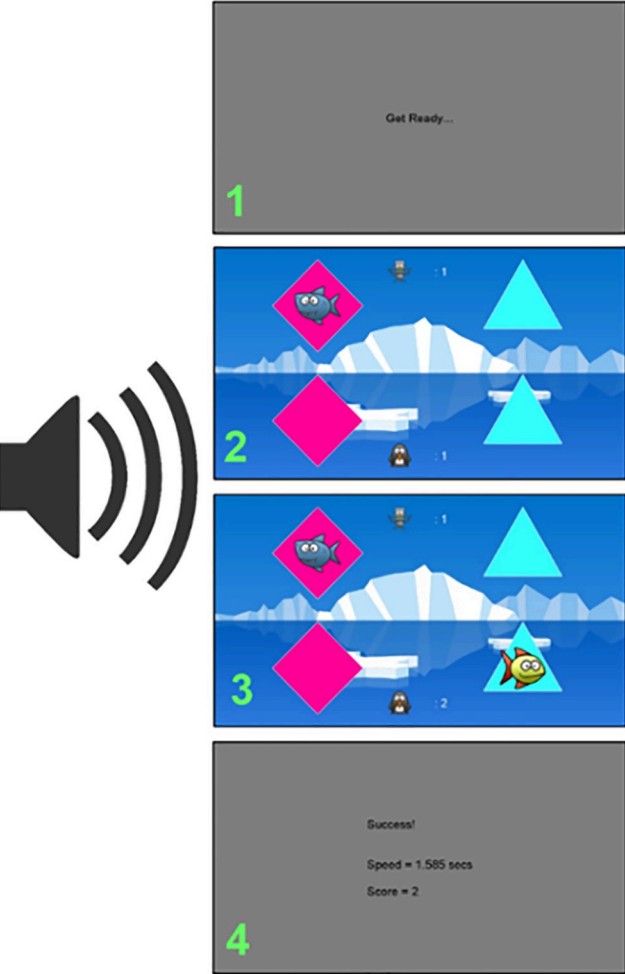

**Fig 1. Example trial of the binary choice task being completed with the secondary task.** This is an example of an unsuccessful information trial (shark is displayed).

S1 and S2 Figs, S1–S3 Files and S1–S14 Data). Participants were always instructed to find the fish and avoid the shark, and were awarded one point on trials in which they correctly selected the stimuli that displayed the fish. Reward location was fixed, so a WSLS strategy was always successful. The task therefore models the most basic requirement for cumulative cultural evolution discussed above (section Testing capacities for cumulative culture). This task is similar to that used by Atkinson et al. [21] in young children. Atkinson et al. found that, although young children could apply a selective strategy, they had a significant bias against repeating information they were shown in an information trial. This study therefore aims to establish whether adult participants in a similar task are still able to overcome this bias to consistently apply a selective strategy, even under additional cognitive load.

*Dual tasks*: *Switching task*. Participants listened to a series of auditory tones through headphones. Tones came in sets of one or two, played at regular intervals. Participants were asked to respond to each tone set with a pre-specified mouse response—clicking the same number of times as the tones heard. After an auditory cue they were required to switch to a different pre-specified mouse response—clicking once more than the number of tones heard. They were instructed not to click in response to the switching cue. For example, if the sequence '1 tone, 2 tones, *switching cue*, 1 tone, 2 tones' was heard, the required mouse response would be '1 click, 2 clicks, no clicks, 2 clicks, 3 clicks'.

*Dual Tasks*: *Control task*. This was identical to the *switching* task, but participants would continue with the same mouse response for the duration of the task. No switch cues were played.

**Procedure.**   Participants were tested individually, although the majority of participants took part in the same room and at the same time as another participant. When two participants took part at once both participants were verbally instructed by the experimenter that they were acting entirely independently and were not competing with one another. Participants were facing away from each other so could not see each other's screens. They were either both taking part in silent tasks or were both wearing headphones so audio distraction from the other participant was minimal. Both participants began the task at the same time and if one participant finished in a quicker time than the other they were escorted quietly out of the testing room to ensure the remaining participant was not disturbed.

Before beginning trials participants received detailed on-screen task instructions which included examples of the audio sounds they would be exposed to and the visual stimuli they were looking for. Participants then completed 4 practise trials of the dual-task on its own, followed by 2 practise trials of the binary choice task. After this brief practise, participants were then required to complete 16 trials of pre-test training in each condition (*switching* and control), with the order of conditions counterbalanced across participants in full dual-task conditions. For the secondary task, the number of trials in each block was determined by the time taken to complete the binary choice task.

To ensure full focus was given to both tasks an inclusion criterion of 75% accuracy was set for both tasks, with a larger participation reward available to participants that completed the full study. Participants that scored below 75% accuracy in the training round (averaged over both blocks) in one or both tasks left the study at this point and received a smaller reward (a reduced participation fee, or fewer research participation tokens, relative to participants completing the full study). Participants were made explicitly aware of this inclusion criterion when signing up to take part in the study. They were also reminded of this again when giving consent to take part, and then once more as part of the written instructions for the study.

Participants who passed the training round then completed a further 72 trials of testing in each block (one EF block and one control block) in full dual-task conditions. For the

secondary tasks the number of trials was determined by the time taken to complete each block of the binary choice task.

Participants were randomly split into 3 equal groups which determined which information trial types they received (the training trials for all participants were set as though they were in the selective copying group):

*WSLS (selective copying group)*. In this group there was a 50% chance of observing a successful or unsuccessful information trial, and whether the computer selected the shape on the left or the right of the screen. There was therefore an equal balance of successful and unsuccessful selections on each side of the screen. These were presented in a fully random order.

*WS (always-copy group)*. In this group 100% of information trials were successful. There was still an equal balance of whether the shape on the left or the right was selected, presented in a fully random order.

*LS (never-copy group)*. In this group 100% of the information trials were unsuccessful. There was still an equal balance of whether the shape on the left or the right was selected, presented in a fully random order.

Overall, 45 participants were excluded based on their training round score (WSLS group: 18, WS group: 16, LS group: 11).

We predicted that reaction times would be slower in the block where participants completed the switching secondary task, compared to the control task block. We also predicted that the impact of the secondary task would be greater in the WSLS, selective copying group than in either of the WS or LS groups. No specific predictions were made about the difference in reaction times between the WS and LS groups.

## Results

**Outliers.**   Overall 1279 outliers were removed from the data for extremely long or extremely quick reaction times. Outlier removal is important for these data in order to screen out responses that were made so quickly they may reflect an accidental button push or repetitive pressing without paying attention to the task, or so slowly that they may be due to external distractions unrelated to the task. Each outlier represents a single trial. A relatively broad inclusion criterion (3 x Median Absolute Deviation (MAD) from the mean; [22]) was used to ensure genuinely long reaction times caused by dual task interference, which were the expected outputs of the study, were not artificially trimmed. Outliers were removed per participant, per block, in line with recommendations from Ratcliff [23] regarding outlier removal for reaction time datasets with high inter-participant variation. 1211 outliers were removed from the upper end of the response distribution and 68 were removed from the bottom end. This represents 7.4% of the total data.

**Analysis: Binary choice task.**   Fig 2 shows the overall reaction time for each strategy group by block condition. Accuracy in the choice task was at ceiling (accuracy range: 98.1%-98.6%). Mean reaction times for each strategy group and block condition are given in Table 1.

Reaction times were analysed using a linear mixed effects model with fixed effects of group (WSLS, LS or WS), block condition (switching or control) and item number (within block), and their interactions. Participant ID was included as a random effect. The WSLS group in the control condition was taken as the baseline and *p*-values were estimated from the resultant t-statistics with degrees of freedom being the number of observations minus the number of fixed parameters in the model [24].

The model was a significantly better fit than the null equivalent ($\chi2(11) = 531$, $p<0.001$). There were significant effects of group membership (LS faster than WSLS: b = -53.7, SE = 22.3, t(15994) = -2.41, $p$ = .032; WS faster than WSLS: b = -77.6, SE = 22.3, t(15994) = -3.48, $p$ =

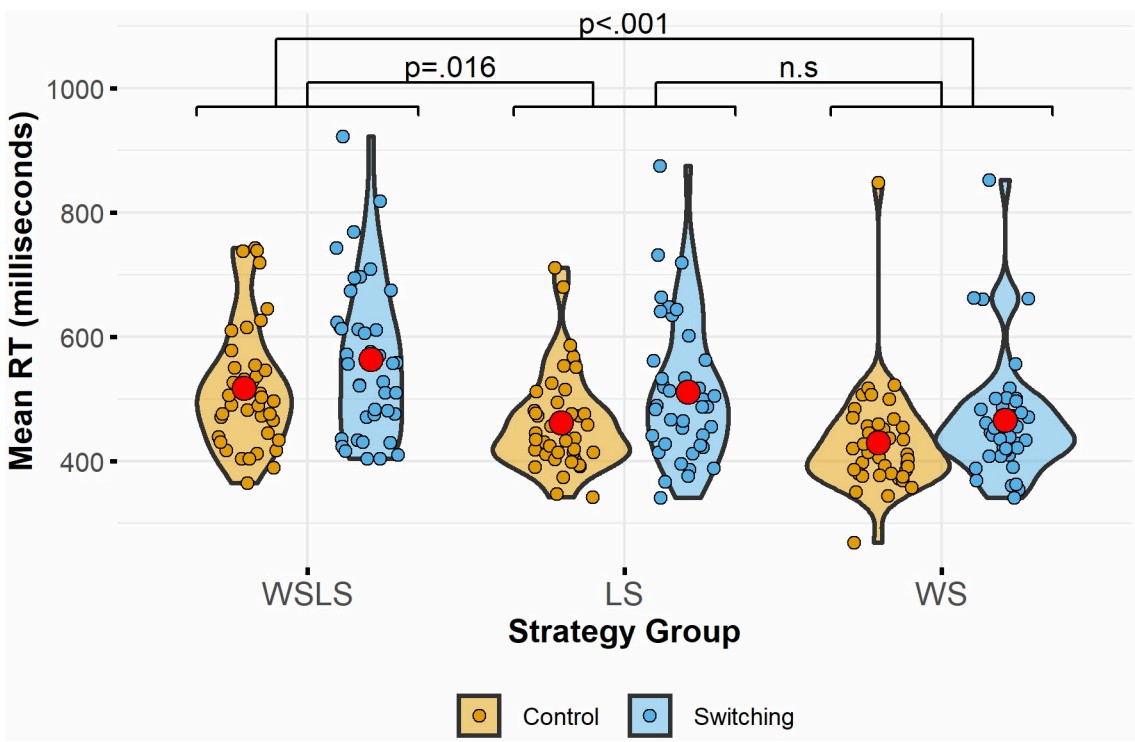

**Fig 2. Reaction Times (RT) in the binary choice task for each group and block condition.** Red dots indicate the mean in each block. Labelled brackets indicate significance level of comparison between strategy groups overall. The overall reaction time in the selective copy group (WSLS) was significantly slower than both the always copy (WS) and never copy (LS) groups. There was no significant difference overall between the LS and WS groups. The difference between the control and EF block condition was not significantly different for each strategy group.

.001; both corrected for multiple comparisons using Bonferroni-Holm correction), secondary task block condition, with the switching block slower than the control: (b = 52.6, SE = 7.80, t(15994) = 6.75, $p < .001$) and item number, with reaction time decreasing as trials increase: (b = -0.438, SE = 0.132, t(15994) = -3.31, $p < .001$). Post-hoc pairwise comparisons of group membership using Bonferroni-Holm correction indicated that the overall difference between the single response-groups, LS and WS, was not significant (b = -23.9, SE = 22.3, z = -1.07, p = .284). The interaction between group membership and block condition was not significant for either single response-type group compared to the selective copying group (LS: p = .483, WS: p = .425). Taken together these results indicate that the selective strategy condition of the binary choice task was more challenging than either of the single-response conditions and, although the *switching* task had a negative impact on response times in each group (compared to the

**Table 1. Mean reaction times (milliseconds, SD in brackets) for each strategy group and block condition.**

| GROUP MEMBERSHIP | BLOCK CONDITION | Mean (sd) reaction time (ms) |
|---|---|---|
| WSLS | Switching | 564 (206) |
|  | Control | 518 (154) |
| LS | Switching | 512 (200) |
|  | Control | 462 (144) |
| WS | Switching | 467 (172) |
|  | Control | 429 (139) |

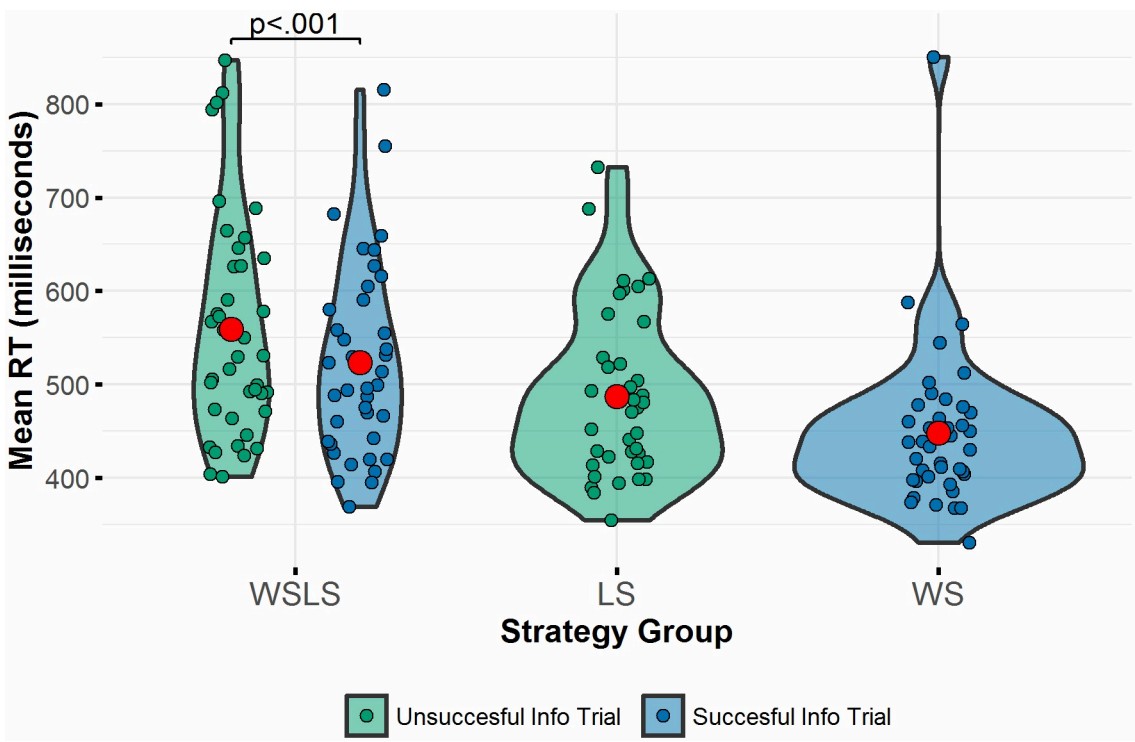

**Fig 3. Reaction times (RT) based on observing successful and unsuccessful information trials.** Red dots show mean in each block or condition. Brackets indicate significance level of difference between blocks.

control task), it did not have a *greater effect* on the selective strategy group, relative to the other groups.

Response time after successful or unsuccessful information trials was analysed using a linear mixed effects model with one fixed effect of success in the information trial and one random effect of participant ID. This model was a significantly better fit than the null equivalent ($\chi2(1)$ = 79.4, p < .001). Taking all groups together, responses were significantly faster after successful information trials (wins) than after unsuccessful information trials (losses) (b = -34.8, SE = 3.9, t(15999) = -8.92, p < .001).

Looking at the selective strategy group only, a linear mixed effects model was constructed with fixed effects of block condition, success in the information trial and their interactions. The control block was taken as the baseline. Participant ID was included as a random effect. This model was significantly better than the null equivalent ($\chi2(3)$ = 204, p < .001). Reaction times were significantly faster after successful than after unsuccessful information trials (b = -35.2, SE = 5.6, t(5384) = -6.29, p < .001) and significantly slower in the executive function compared to the control block (b = 45.8, SE = 5.70, t(5384) = 8.03, p < .001). There was no significant effect of the interaction between block condition and information trial success (p = .975).

Fig 3 shows mean reaction times by information trial type.

**Analysis: Secondary task.** Participant accuracy was at ceiling in the control condition, and significantly above chance in the executive function condition: performance in each block of each condition was significantly above a chance level of 50% as shown by binomial testing (p < .001 for all conditions). Statistical analysis of the secondary task is given in the S1 and S2 Figs, S1–S3 Files and S1–S14 Data.

## Discussion

Overall there was a significant effect of the secondary task on the binary choice task, with reaction times in switching blocks significantly slower than in the control blocks. However, the impact of the secondary task was the same across all group conditions, with no significant differences in how much the concurrent task slowed down performance.

This may indicate that there is an executive function requirement even in such a simple task, causing performance disruption even in conditions where it was not predicted. Even though the tasks were intended to be very simple for participants, the impact of the switching task may show that even basic decision-making draws on system-2 resources.

However, given the same level of interference was found across conditions it may also indicate that, although the WSLS condition required less predictable responses than the other conditions, all conditions were extremely easy for participants. Ceiling effects may therefore have obscured any potential differences between conditions in the impact of the switching task relative to the control task. The significant effect of item number on reaction times suggests a training effect in the main task and thus supports this conclusion, as it could indicate that participants reached ceiling levels of performance in the main task before the end of each block. The decrease in reaction times also suggests that responses to the main task were becoming increasingly well-practiced, which is likely to decrease dependence on system-2 resources with greater task experience. As the reaction times in the choice task decreased, accuracy in the concurrent audio *switching* task also decreased (see S1 and S2 Figs, S1–S3 Files and S1–S14 Data). This may suggest some offloading of task demands onto the concurrent task. However, the amount of offloading appears to remain the same across strategy groups.

Although there was no difference in how much the secondary task affected each strategy group, there was a significant difference between overall task performance in the three different conditions. Reaction times in the WSLS group, the only condition requiring behaviour analogous to an explicit, system-2 social learning strategy, were significantly slower than those in either of the non-selective conditions. Responses were also significantly faster after successful information trials (wins) than unsuccessful information trials (losses), overall and in the WSLS condition. This indicates that participants found responses after 'wins' easier than responses after 'losses', which would suggest that the lose-shift condition should be the most challenging, as it is comprised only of unsuccessful information trials. The finding that the flexible strategy was slowest therefore indicates that flexible strategy use is significantly more challenging than applying a simple additive combination of each component action within the strategy; the flexible nature of the strategy itself is what makes it challenging.

## Experiment 2

In order to establish whether true differences in dual task interference between using a selective copying strategy and repeatedly applying a consistent rule were masked by ceiling effects in the task, further data were collected for experiment 2 using a task that was more challenging for participants. The more challenging task followed the same basic structure but now two selections, out of four possible options, were made on the information and test trials, rather than one out of two. The aim of this was to increase the requirement to make selective choices, in the selective strategy condition, as on some trials two different strategies (both WS and LS) would be required at once in order to perform optimally at the task. The strategy requirements in the repeated copying (WS group from E1) condition, however, would remain the same.

The simultaneous application of two information use strategies (retain useful elements and deviate from detrimental elements of a single demonstration), increases the ecological validity of the task, as it models more realistically the cognitive requirements of learning situations that

may lead to cumulative culture. For example, if observing a conspecific foraging successfully in some locations and unsuccessfully in others, selective copying of only the successful locations combined with deviating from the unsuccessful locations would be required in order to out-perform the observed model. In copying situations such as this it is unlikely that the observer would be able to act simultaneously with the model (due to e.g. social status, availability of for-aging locations or tools). Consequently, there will also be some memory requirement involved to retain information about what should be copied and what should be deviated from. An additional memory load condition was therefore also introduced, to increase the overall diffi-culty and ecological validity of the task. This was added for both the selective strategy condi-tion and the repeated copying condition.

A direct measure of metacognition was also tested in E2. This enabled a direct test of whether the aim of restricting access to participants' metacognitive reasoning, as outlined in section Dual-systems and dual-tasks, was indeed fulfilled by the secondary switching task. It also provided a measure of metacognitive efficiency that could be correlated with proficiency in using a selective copying strategy; we predicted this relationship to be positive.

## Methods

**Participants.** Participants were recruited at the University of Stirling and took part either in exchange for course credit or cash remuneration at a rate of £5/hour rounded up to the nearest pound. One-hundred and fifty-one participants took part (39 male, mean age: 20.4, age range: 17–35). 57 participants completed the training phase of the experiment but did not complete the full testing phase (see S1 and S2 Figs, S1–S3 Files and S1–S14 Data for a break-down of exclusions and accuracy thresholds required to pass all tasks). In total 94 participants (23 male, mean age: 20.2, range: 17–35) completed the full study and were included in the dataset reported in the analyses below. All participants had normal or corrected to normal vision and hearing. Participants all gave written consent to take part and were aware that they were free to withdraw from the study at any time. All participants were task naïve, as partici-pants that took part in the pilot or E1 were not able to sign up to E2. Ethical approval for the study was given by the University of Stirling General University Ethics Panel (reference GUEP 468). Participants aged 16–18 years old were treated as adults and gave their own consent, in line with British Psychology Society guidelines.

**Apparatus.** Participants were tested using a desktop computer running Windows 8 with a standard mouse, a Black Box Toolkit 4 button response pad button box and Goji over-ear active noise cancelling headphones. All tasks were written and run in PsychoPy version 1.84.2 [20].

**Procedure.** The procedure remained very similar to that used in Experiment 1 with changes described below. All participants completed both the choice task and the metacogni-tion task described below. Whether participants completed the choice task or the metacogni-tion task first was counterbalanced across participants. Both tasks were completed alongside the concurrent *switching* dual-task and control dual-task used in Experiment 1. The order of the switching and control blocks was again counterbalanced across participants. Due to the low pass rate in Experiment 1, a minor change was made to the switching task to lower the memory requirements. The requirement of the participant to keep track of previous switching behaviour was removed. The task now had two different switch cues: one to indicate that participants should use each of the two pre-specified mouse responses (see S1 and S2 Figs, S1–S3 Files and S1–S14 Data for more details). As Experiment 2 was longer than the previous two experiments, five-minute breaks were added between the training round and first test round, and between the first and second blocks of testing to ensure participants did not experience task fatigue.

*Choice task*. The procedure used was similar to that used for the binary choice task in experiment 1. However, the number of conditions was reduced so a comparison was only made between a selective copy and an always copy condition. The difficulty of the task was increased so that participants were now required to make two selections out of four rather than one out of two. This meant that in the selective copy condition there was also more variability of information trial types to include an equal balance of trials which revealed two successful stimuli, two unsuccessful stimuli, and one successful and one unsuccessful stimulus. This condition therefore included the only trials in which a participant could not automatically deduce every correct reward location: on trials that showed only one successful stimulus there was a choice of two locations which could contain the reward. In those trials, the likelihood of the reward being revealed in the selected location was set to 50%. In the always copy condition the information trial would always show two successful stimuli. Participants completed 18 trials in the training phase followed by 54 trials in each block of the experimental phase.

A memory load was also included for 50% of participants in each strategy group. With the memory load present the information trial would not remain visible while the test trial selections were being made. A time limit was also added to all trials, so if the first response was not made within three seconds the trial would time out and one point would be removed from the participant's total score. See Fig 4 for an example of the procedure.

*Metacognition task*. The metacognition task assessed metacognitive monitoring ability, and consisted of a two alternative forced choice (2AFC) task in which participants were required to decide which of two patches of white dots was sparser (see Fig 5). They were then asked to rate their confidence in that decision on a scale from 1–4 (1 being least confident, 4 being most confident), using the 4 buttons of the button box. Points were gained for correct responses and lost for incorrect responses, with the number of points gained or lost equal to the confidence rating given. For example, a correct response rated with confidence '3' would score 3 points, and an incorrect response rated with confidence '2' would lose 2 points. This symmetrical points structure was used to try to motivate participants to avoid rating responses they were unsure of with high confidence, due to the risk of losing a high number of points (although see *Discussion* for the potential negative impact of this scoring system). Immediate feedback about the number of points gained or lost, and therefore the correctness of the response, was given after each trial and a running total of points was given in the top right corner of the screen throughout the experiment. This running score could be negative. A selection needed to be made within five seconds or the trial would time out and one point would be taken from the total. No time limit was imposed on the confidence rating. The metacognition task was completed concurrently with the switching dual-task and control dual-task.

There were 7 difficulty levels, with each level twice as hard as the level before it (set by percentage difference in the density of the patch, see SI for full details). This was to ensure some trials could definitely be answered correctly and some would have to be answered based on a random guess between the two stimuli, to ensure a mixture of correct and incorrect answers and to encourage use of the full scale of confidence ratings. The difficulty levels were set during piloting to aim for an estimated average accuracy of around 75% in the task. The same difficulty levels were used across participants. Participants completed 28 training trials and 56 test trials, with an equal number of trials at each difficulty level, presented in a random order.

*Experiment two predictions*. We predicted that, as in experiment 1, the secondary switching task would make reaction times in the choice task slower. We again predicted more dual-task interference in the WSLS group than the WS group, due to the requirement to use a selective strategy in the WSLS group. This is in line with the EMCC prediction that the use of selective strategies relies on explicit processes. We did not make specific predictions regarding the addition of a memory load to the choice task. The EMCC would predict that a task that reduced

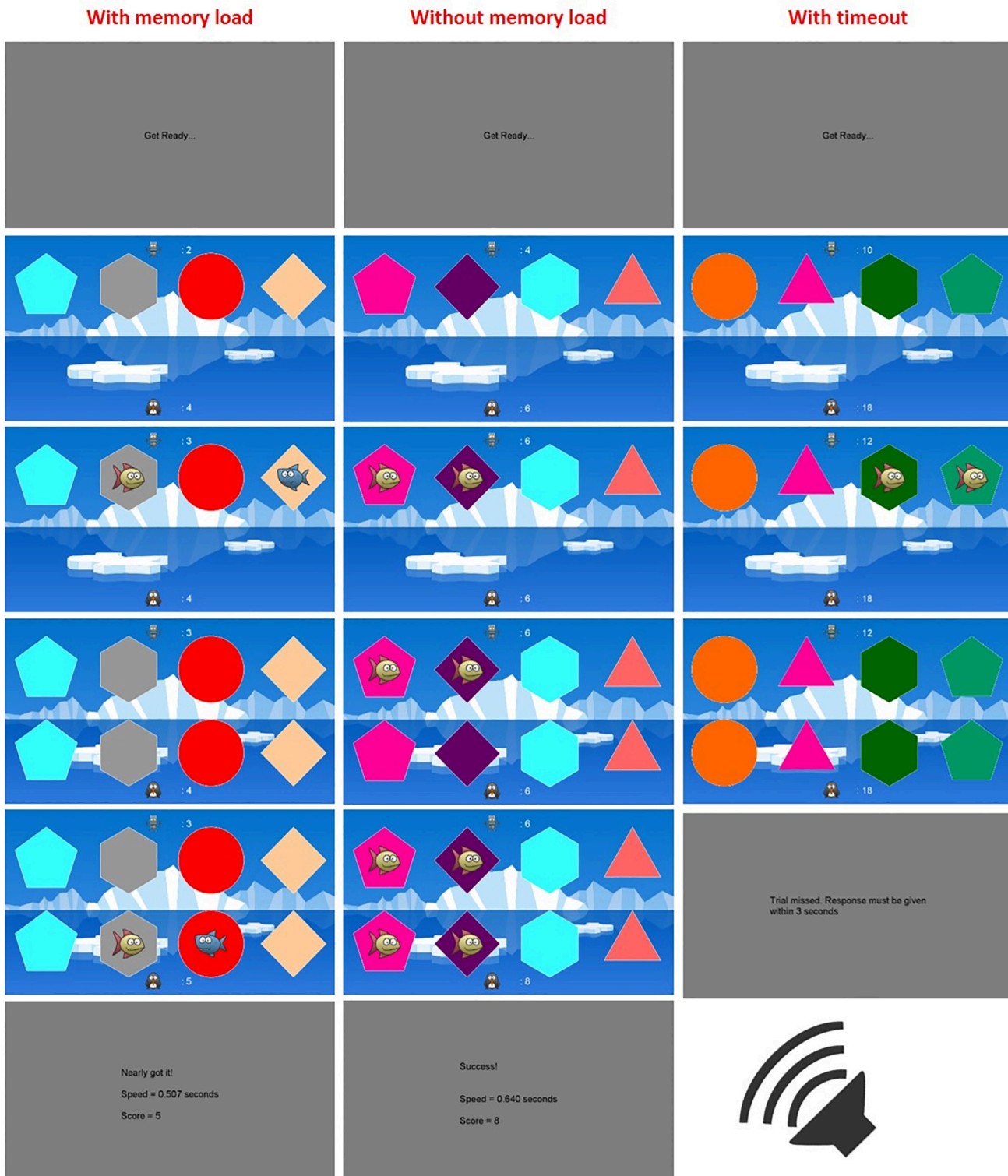

**Fig 4. Example trials of the choice task.** In all conditions the task was completed alongside the audio switching task. **Left:** an information trial with one succesful (fish) and one unsuccesful (shark) stimuli, with an additional memory load. The test trial shows the participant has used a correct strategy and repeated the successful information, and shifted away from the unsuccessful information, although due to chance they have selected one unrewarded stimulus. **Middle:** an information trial with two successful stimuli, with no memory load. The test trial shows two successful stimuli are selected. **Right:** a trial with a timeout for no response.

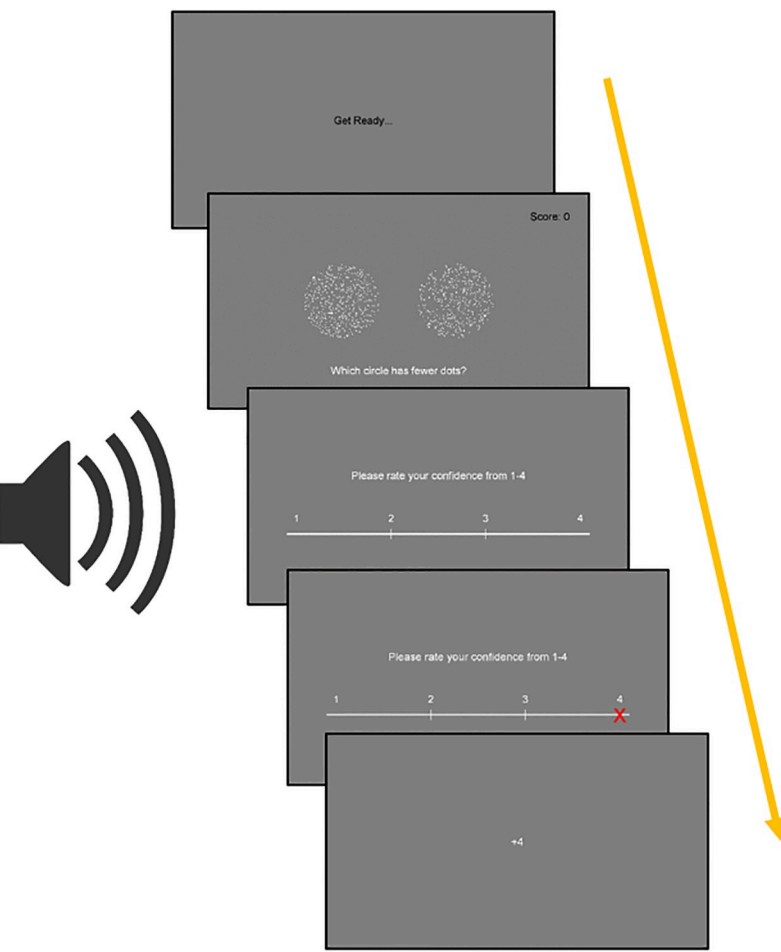

**Fig 5. An example of a trial of the metacognition task.**

capacities for selective copying would also have a negative impact on metacognitive efficiency. We therefore predicted the *switching* dual-task would have a negative impact on metacognitive efficiency in the metacognition task, in line with the predictions for the choice task.

## Results

**Choice task outliers.** Overall 531 outliers were removed from the data for very long or very short reaction times, using the same procedure for outlier removal as in Experiment 1. Four-hundred-and-seventy-three outliers were removed from the upper end of the response distribution and 58 were removed from the bottom end. This represents 5.2% of the total data. Outliers have only been removed from participants' first responses. This is because some second responses were extremely quick as participants were using a response strategy of deciding on both of their selections before making either and then pressing both buttons almost simultaneously, so removing these trials would create an incorrect representation of the data. If R1 was removed on any particular trial, the entire trial was removed from the dataset so R2 was also removed.

**Analysis.** Table 2 shows the mean reaction times in each block and condition.

Correct WSLS strategy adoption was at ceiling with correct strategy use nearly 100% of the time across conditions (range: 99.5%-100%).

**Table 2. Mean reaction times at R1 (first response) and R2 (second response) of choice task.**

| GROUP MEMBERSHIP | BLOCK CONDITION | MEMORY LOAD CONDITION | Mean (sd) R1 (ms) | Mean (sd) r2 (ms) |
|---|---|---|---|---|
| WSLS | Switching | Memory Load | 341(228) | 225(272) |
| | Control | Memory Load | 307(186) | 221(237) |
| WS | Switching | Memory Load | 335(185) | 251(189) |
| | Control | Memory Load | 326(176) | 230(189) |
| WSLS | Switching | No Memory Load | 307(220) | 309(238) |
| | Control | No Memory Load | 303(205) | 305(275) |
| WS | Switching | No Memory Load | 331(216) | 191(246) |
| | Control | No Memory Load | 314(186) | 185(228) |

Fig 6 shows the reaction times at R1 and R2, split by block condition, memory load and group.

Response times were analysed using a linear mixed effects model with fixed effects of group (WSLS or WS), block condition (secondary switching task or secondary control task), memory load, response number (first (R1) or second (R2) response), item number, level of success of the information trial (whether it scored 0, 1 or 2 points) and the interactions between group, block condition, memory load and response number. Participant ID was included as a random effect. The first response (R1) of the WSLS group, in the control condition with no memory load was taken as the baseline and *p*-values were estimated from the resultant t-statistics with degrees of freedom being the number of observations minus the number of fixed parameters in the model [24]. R2 is measured from R1 rather than stimulus onset.

The model was significantly better that its null equivalent ($\chi 2(17) = 1123$, $p < .001$).

The EMCC would predict that the dual-task would have more of an impact in the WSLS group, so would therefore predict a greater performance impairment in the switching block (relative to the control block) for the selective (WSLS) group than for the WS group. Longer reaction times would also be predicted when under an additional working memory load. These predictions are supported by the significant interaction between group, block condition and memory load ($b = -35.8$, SE = 16.5, $t(19247) = -2.17$, p = .030) (see S1 Fig). This interaction relies partially on the increased impact of the additional memory load on the WS group, which is not predicted by the EMCC. However, the differences in overall reaction time are likely affected by the changes between R1 and R2 which may rely on a planned metacognitive strategy (see D*iscussion*).

There was an additional significant main effect of item number, indicating faster responses as trial number increased: ($b = -0.787$, SE = 0.094, $t(19247) = -8.38$, p < .001).

There were also multiple significant interactions. For clarity, relevant interactions are described in detail below with some additional interactions described in the SI. Post-hoc comparisons of the interactions were carried out using estimated marginal means (using the emmeans package in R).

*Group, block condition and memory load (see above).* In the WSLS group, both block conditions showed a reaction time decrease when a memory load was present, compared to there being no memory load. This was more pronounced in the control block. Conversely, in the WS group both blocks showed a reaction time *increase* with a memory load present (see S5 Fig).

*Block condition and memory load (b = 29.5, SE = 11.5, t(19247) = 2.56, p = .011).* When there was no memory load the difference between the executive function and control blocks approached significance (b = -7.33, SE = 4.14, t(19157) = -1.77, p = .077), but when a memory load was present the switching block was significantly slower than the control block (b = -17.5,

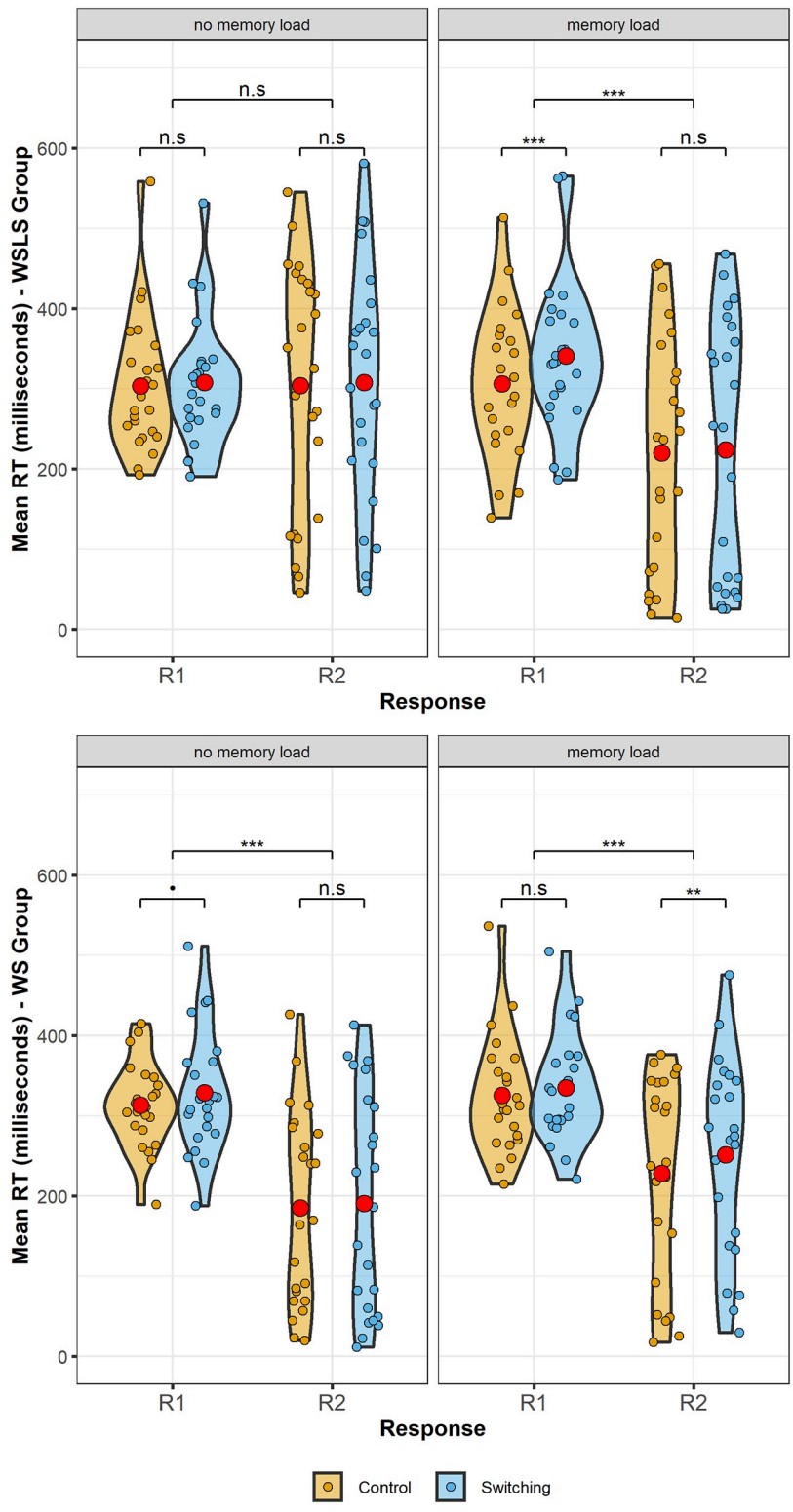

**Fig 6. Reaction times at R1 and R2 for the WSLS group (top) and WS group (bottom), split by block condition and memory load.**

SE = 4.13, t(19156) = -4.25, p<0.001). This indicates that the secondary task had a significant impact on reaction times only when combined with the additional memory load.

*Group*, *memory load and response number* (b = 120, SE = 16.5, t(19247) = 7.28, p < .001), and *group*, *block condition*, *memory load and response number* (b = 52.1, SE = 23.4, t(19247) = 2.23, p = .026):

In the WS group there was a significant decrease in reaction time between R1 and R2, both with and without a memory load and in both blocks (p < .001 for both). This decrease was also significant when split between the switching and control blocks (again p < .001 for both blocks). However, in the WSLS group this reaction time decrease only applied when a memory load was present; when there was no memory load there was no decrease in reaction time (R1-R2, overall: b = -1.42, SE = 5.76, t(19156) = -0.246, p = .805; control block: b = -1.74, SE = 8.13, t(19156) = -0.214, p = 0.803; R1-R2, switching block: b = -1.10, SE = 8.16, t(19156) = -0.314, p = .893) (see S2 Fig).

Overall the interactions indicate that although the secondary switching task had a similar effect on the different strategy groups, the addition of a memory load significantly changed the way participants behaved in the different strategy groups. Across both strategy groups the detrimental impact of the secondary task on reaction times was also contingent on the memory load being present.

**Metacognition task.**    Mean accuracy in the visual discrimination of the stimuli in the metacognition task was approximately 76% (control block: 75.9%; switching block: 76.3%), close to the desired accuracy based on the difficulty levels set during piloting. However, this mean accuracy level reflects a wide range of accuracy, with some participants performing much higher than expected and some participants performing much lower (control range: 62.3–87.5%; switching range: 64.3–91.1%). A repeated measures ANOVA showed no significant difference in discrimination accuracy between control and switching blocks (F(93) = 0.291, p = .591).

Participants in both blocks tended to rate their confidence highly, with an aversion to using the lowest confidence level even at the hardest difficulty level (see Fig 7).

Success on each trial was analysed using a binomial general linear mixed effects model with fixed effects of block condition, scaled confidence and scaled difficulty level and their interactions. Participant ID was included as a random variable and the control block was taken as the baseline. The model was significantly better than the null equivalent (x2(7) = 2329, p < .001). There were significant effects of confidence level (b = 4.45, SE = 0.395, z = 11.3, p < .001) and the interaction between confidence and difficulty level (b = -2.50, SE = 0.293, z = -8.56, p < .001). This indicates increased success when participants were more confident, although the increase does not apply equally to all difficulty levels (see Fig 8). This increase is particularly pronounced from confidence level 1 to level 2, although this may be largely due to the low number of responses given with confidence level 1.

The impact of the secondary task on participants' metacognition was analysed using the metaSDT package in R [25]. A single score of metacognitive efficiency (m-ratio) for each participant in each block was calculated as metacognitive sensitivity (meta-d'$_b$: balanced model fit) divided by overall sensitivity in the visual task (d'). Meta-d' is defined as the type-I sensitivity (accuracy in the visual discrimination task) that would be found if all of a participant's type-II ratings (subsequent confidence ratings) were considered to be optimal [26]. This measure aims to give a bias free measure of metacognition which is not affected by performance in the visual task or a bias towards over- or under-confidence [27]. A repeated measures ANOVA showed no significant difference in m-ratio between control and switching blocks (mean control: 1.13, mean switching: 1.03, F(93) = 0.663, p = .418) (see Fig 9).

**Correlation between choice task and metacognition task.**    The mean difference in reaction time between the switching and control blocks of the choice task was calculated for each

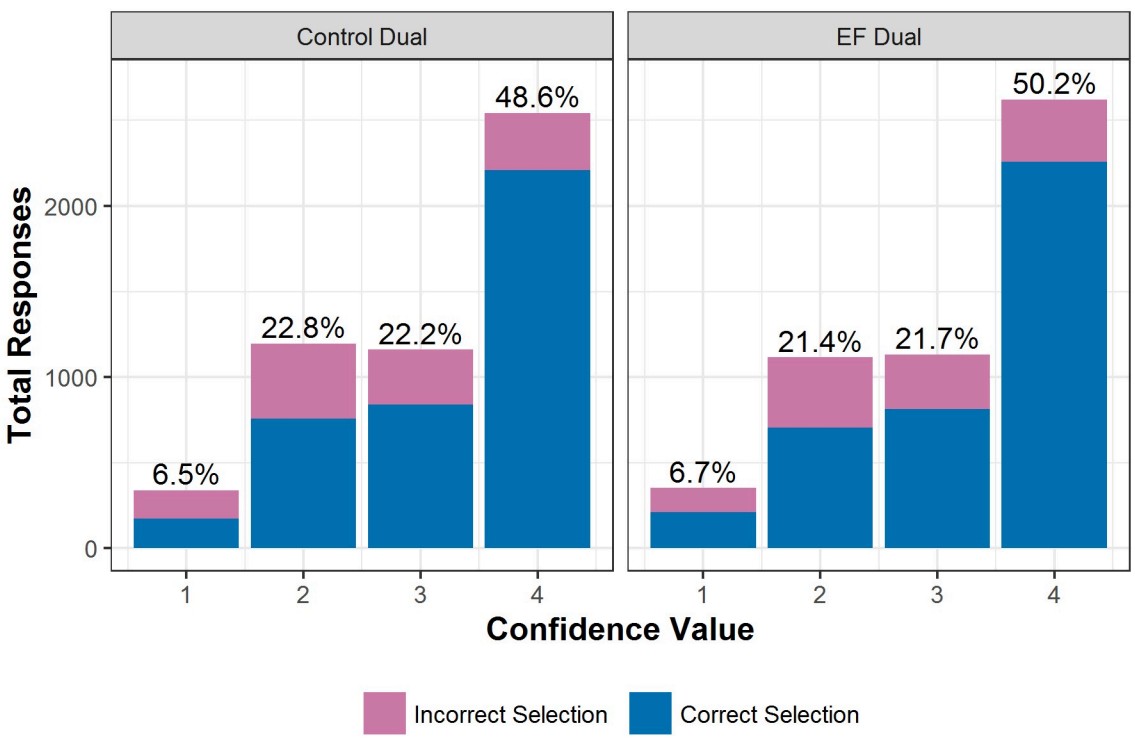

**Fig 7. Total responses given at each level of the confidence scale, split by the accuracy of the visual discrimination.** Percentages indicate the percentage of all responses in that block at that confidence level.

participant. This reaction time difference reflected the impact of the dual-task. Overall, there was no correlation between dual-task impact and baseline metacognitive efficiency (taken as m-ratio for the control block only) for either R1 or R2 (R1: r = .011, t(92) = 0.102, p = .919; R2: r = .070, t(92) = 0.675, p = .501), indicating no relationship between greater metacognitive efficiency and impact of the secondary task on speed of applying a flexible copying strategy. This may indicate that, in contrast with the predictions of the EMCC, increased metacognitive monitoring accuracy does not necessarily lead to an increased proficiency in applying a flexible strategy.

**Secondary task.** Participant accuracy in the audio switching task was at or close to ceiling in all conditions; accuracy range 86.5%-95.7%. Statistical analysis of the secondary task is given in the S1 and S2 Figs, S1–S3 Files and S1–S14 Data.

## Discussion

The EMCC predicts that reduced access to executive resources should have a negative impact on capacities for cumulative cultural evolution, due to a reduced ability to make the decisions that form the building blocks of CCE–namely to improve upon observed information by making deliberate use of selective copying strategies. It would therefore predict less efficient (either slower or less accurate, or both) strategy use when participants are under a dual-task load.

Accuracy in strategy adoption was at ceiling throughout the task. The data show no overall effect of the secondary task on response times in the choice task, despite the choice task being more challenging than the binary task used in E1. There was also no significant difference found between groups overall. However, there was a significant difference between the groups

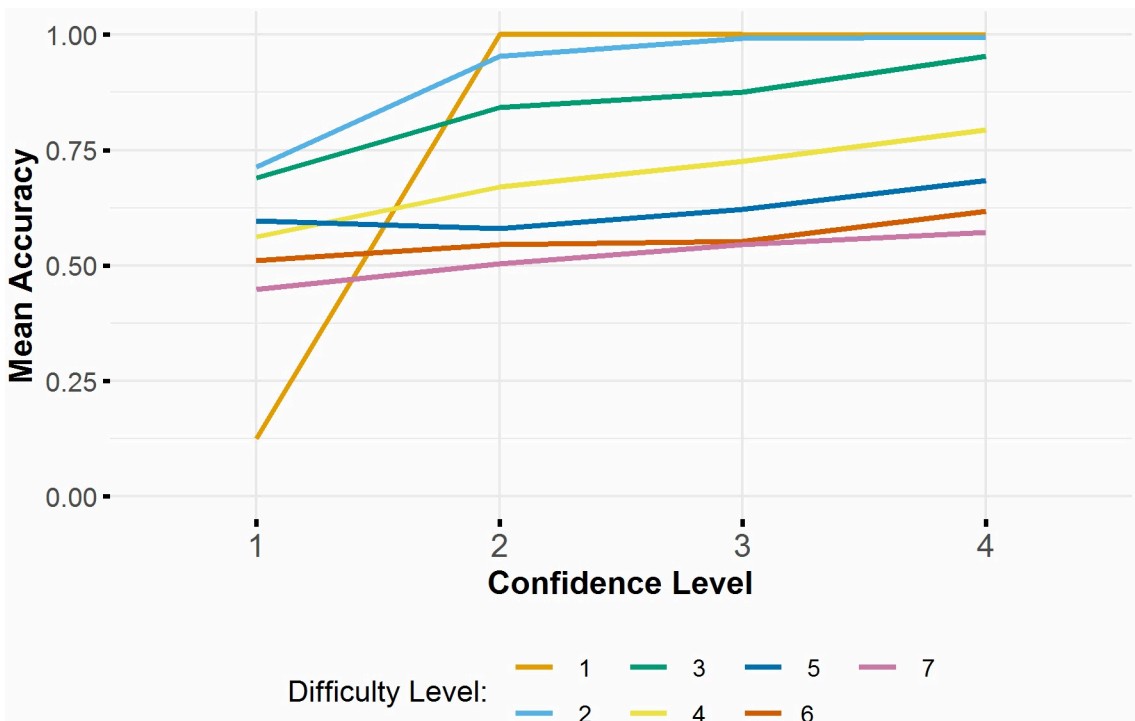

**Fig 8. Accuracy in metacognition task at each confidence level, split by trial difficulty.** Difficulty Level 1, least difficult; Difficulty Level 7, most difficult.

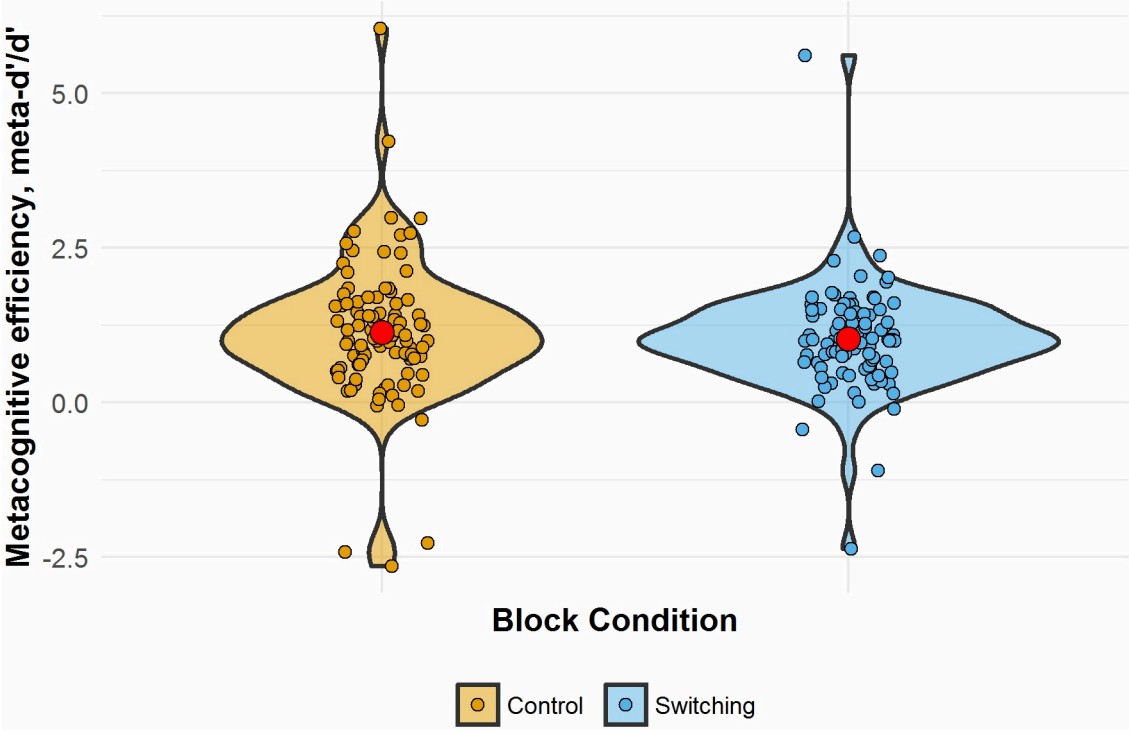

**Fig 9. M-ratio (metacognitive efficiency) for the control and EF blocks.** Red dots indicate the mean for each block.

in terms of how much the memory load affected reaction times. The WS, always-copy group appeared to be relatively unaffected by the additional memory load: although there was an overall increase in reaction times with an additional memory load this was not significant and the difference between R1 and R2 remained the same both with and without a memory load. In contrast, the WSLS, selective-copy group showed a significant increase in reaction time in the switching block at R1, followed by a significant overall decrease in reaction time from R1 to R2, when the memory load was applied.

This may indicate that participants were aware that the information trial would not remain visible when making their selections, and so R1 and R2 are planned before making R1. This then leads to a very quick R2, as participants are able to select both stimuli almost simultaneously. The increased reaction time in the switching-block only, indicates an executive function cost to this 'planning to remember'. When no memory load is present this planning is not required, so there is no reaction time increase at R1 but also no decrease at R2. This response strategy is in itself metacognitive, as it involves monitoring and control of the response behaviour. This may indicate some involvement of metacognition in efficiently applying flexible copying strategies, and in overcoming the challenges posed by working memory loads.

The finding that the always-copy condition shows a consistent reaction time decrease between responses, whereas the selective-copy condition reaction times are somewhat dependent on the availability of working-memory, mirrors the findings from Experiment 1 that indicate that selective copying is more challenging than always or never copying. This increased difficulty of applying a strategy may rely more on working-memory than task-switching executive resources.

The data from the metacognition task appear to show no difference in metacognitive accuracy when under additional executive function load. There was also no correlation between metacognitive efficiency in the control block and the impact of the dual-task on the choice task. These findings are in contrast to the predictions of the EMCC, and suggest limited involvement of metacognitive monitoring in successfully using the type of flexible copying strategy hypothesised to be requisite for CCE. However, the outcome of the metacognition task may be impacted by the tendency of participants to make much more use of the highest confidence level than might be expected if participants were responding with honest confidence ratings. Although the meta-d' method of assessing metacognitive efficiency is designed to be robust against type-II response bias, the measure can be affected by very high or low values of type-II hit and false alarm rates [25]. The tendency to only respond with the higher end of the confidence scale may also have been exacerbated by the payoff structure in the metacognitive rating task: with a symmetrical points payoff structure, overall payoff in the task will tend to be higher (assuming an average success rate higher than chance levels of 50%) if a high rating is always given for confidence. This may have allowed participants with low metacognitive sensitivity to still score highly by applying a blanket rule of 'select high confidence'.

The m-ratio scores may also have been artificially inflated due to the task design that had multiple varying difficulty levels. This design had the desired effect of ensuring some trials were much more difficult than others and therefore all participants had a combination of correct and incorrect answers. However, Rahnev and Fleming [28] suggest that staircase procedures or tasks with multiple different difficulty levels can exaggerate estimates of metacognitive efficiency, as there can be direct comparison between trials which are definitely very hard with trials that are definitely very easy. This could be alleviated in future testing by using a single, fixed difficulty level and awarding points for metacognitive accuracy in a different way, for example by using a strictly proper scoring rule. A strictly proper scoring rule assigns points not simply for accuracy, but for the combination of accuracy and confidence. A high score can therefore be obtained by correct responses rated with high confidence, but also by incorrect

responses rated with low confidence. Similarly, low scores would be assigned on correct trials rated with low confidence or incorrect trials rated with high confidence.

There was a significant decrease in secondary task accuracy in the switching blocks compared to the control blocks for both main task conditions, especially the metacognition task (see S1 and S2 Figs, S1–S3 Files and S1–S14 Data). This may indicate participants were offloading some of the cognitive demands of the switching block to the concurrent audio distractor rather than the main task, with more offloading during the metacognition task than the choice task. This suggests that there is executive function involvement in both main tasks, but the offloading has masked some of the differences between the switching and control blocks.

The above factors mean that definitive conclusions about the relationship between executive processes and metacognitive monitoring efficiency cannot be drawn at this stage. However, this experiment did find some evidence for system-2 involvement in applying flexible copying strategies rapidly due to the effects of an additional working memory load, and the use of a metacognitive *control* strategy was implicated. Future testing could investigate the involvement of explicit metacognition in using flexible copying strategies by amending the metacognition task with the changes noted above, employing a task to measure impact on metacognitive control strategies directly, and ensuring equal or higher motivation was given to the ongoing task to avoid offloading of cognitive costs.

These results do not offer direct support for the EMCC in the ways predicted, as the dual-task manipulation had similar effects on both selective (WSLS) and non-selective (WS) strategies and the data from the metacognition task are inconclusive. However, the differences in reaction times under different memory loads do indicate that metacognitive control may play a role in using selective strategies. This may indicate that, even if metacognitive response strategies are employed, the individual learning decisions making up CCE can occur without explicit access to *task switching* resources. Memory capacity, rather than explicit application of a strategy may play a bigger role in efficient selective strategy use. However, the involvement of working memory resources still implicates the use of system-2 cognition. Whether these findings would hold for more challenging individual copying decisions cannot be confirmed from these data.

## Conclusion

The hypothesis presented at the outset of the study, that system-2 executive resources and explicit metacognition are required to make the learning decisions needed for cumulative cultural evolution to occur (the EMCC), was tested by applying a concurrent executive load to participants completing a visual choice task (E1 & E2). A visual discrimination task followed by a metacognitive judgement was also tested in conjunction with an executive dual-task (E2).

The results indicated that dual-task methods that put additional load on the *switching* executive function effectively restricted access to the cognitive resources required to make rapid, accurate decisions in a simplified social-learning context. However, the switching task had a significant impact on very simple decision making that did *not* require flexible strategy use, as well as on the application of a flexible strategy. These flexible strategies were shown to be more challenging for learners, due to longer reaction times in E1, and fewer reaction time decreases in E2, which may or may not be due to higher executive function demands. The *switching* dual-task used in E2 removed the working memory demands that were present in E1 by removing the requirement for participants to keep track of previous switch cues. Given that the dual-task in E2 did not generate an overall reaction time increase in the choice task, this suggests that the *switching* interference found in E1 may have been caused by working memory demands, rather than switching demands. This is supported by the significant memory load

interference in E2. The different impact of the memory load on the different strategy groups in E2 also indicates system-2 involvement in making selective copying decisions, as a reliance on working memory is a hallmark of system-2 in most dual-systems theories (Evans and Stanovich, 2013).

The results from the E2 metacognition task indicated that dual-task interference did not impair explicit metacognitive judgements. Due to confounding factors of task design, these results are not conclusive. However, some evidence was found for the use of metacognitive monitoring and control strategies when copying flexibly and overcoming additional working memory loads in the E2 choice task.

The results summarised above offer partial support to the EMCC. While executive function interference with explicit metacognitive efficiency was limited, this may reflect the uneven use of confidence levels and inflated metacognition scores within the metacognition task, rather than a genuine absence of dual-task interference. Competition for system-2 executive function and working memory resources did have a detrimental impact on the ability to complete the choice task. However, in some contexts this dual task interference was not limited to the flexible strategy condition. Evidence was also found for the involvement of metacognitive control strategies in overcoming additional memory loads to use flexible strategies efficiently. This implicates the role of system-2 in making ecologically valid copying decisions, which may be beneficial for cumulative cultural evolution. There was evidence of offloading of cognitive demands for both main tasks, also indicating system-2 involvement which may have been masked by unequal focus being given to the main and concurrent tasks. Future testing should be able to ascertain whether this system-2 involvement is in fact a consequence of explicit metacognition.

## Supporting information

**S1 Fig.**
(TIF)

**S2 Fig.**
(TIF)

**S1 Data.**
(R)

**S2 Data.**
(R)

**S3 Data.**
(R)

**S4 Data.**
(CSV)

**S5 Data.**
(CSV)

**S6 Data.**
(TXT)

**S7 Data.**
(TXT)

**S8 Data.**
(TXT)

**S9 Data.**
(TXT)

**S10 Data.**
(TXT)

**S11 Data.**
(TXT)

**S12 Data.**
(TXT)

**S13 Data.**
(TXT)

**S14 Data.**
(TXT)

**S1 File.**
(DOCX)

**S2 File.**
(DOCX)

**S3 File.**
(DOCX)

## Author Contributions

**Conceptualization:** Juliet Dunstone, Christine A. Caldwell.

**Formal analysis:** Juliet Dunstone, Mark Atkinson, Elizabeth Renner.

**Investigation:** Juliet Dunstone.

**Methodology:** Juliet Dunstone, Christine A. Caldwell.

**Project administration:** Juliet Dunstone.

**Resources:** Juliet Dunstone, Mark Atkinson.

**Supervision:** Mark Atkinson, Catherine Grainger, Christine A. Caldwell.

**Visualization:** Juliet Dunstone.

**Writing – original draft:** Juliet Dunstone.

**Writing – review & editing:** Juliet Dunstone, Mark Atkinson, Catherine Grainger, Elizabeth Renner, Christine A. Caldwell.

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
