## [Decision Letter · Decision Letter 0]

21 Oct 2020

PONE-D-20-26673

Flexible social learning strategies are harder than the sum of their parts

PLOS ONE

Dear Dr. Robertstone,

Thank you for submitting your manuscript to PLOS ONE. I have received reviews of your manuscript from two experts in the field, and I am very grateful to both reviewers for providing such thoughtful and clear reviews. I have also read your article myself. The reviewers and I found much to like in your manuscript: the study makes a novel and interesting contribution, and the experiments are well designed. However, both reviewers noted concerns regarding various aspects of the article that would benefit from being  addressed by appropriate revision. Therefore, I invite you to submit a revised version of the manuscript that addresses the points raised during the review process.

I will not reiterate all of the points made by the reviewers since the reviews they have provided are clear and well-argued. It will be important for you to address all of these points in any revision. I will merely highlight some of the key issues here, and add thoughts from my own reading of the paper.

1.  Both reviewers note that the manuscript feels over-long, and I agree with them: as it stands there is a danger of the relatively straightforward message of the paper becoming lost in a mass of methodological detail and statistical analysis. I would therefore encourage you to streamline the article wherever possible. For example, both reviewers suggest moving the details of the pilot experiment to supplementary materials (Reviewer 1 suggests also moving Experiment 1 - I will leave that decision up to you). Likewise it felt like the results sections could be condensed: at the moment it sometimes feels a little like every possible contrast of every dependent variable is being reported, and I found myself getting lost in a sea of statistics. Again I encourage you to strip back the reported analyses to what you consider the most critical aspects that address the hypotheses that you have developed in the introduction (moving less focal analyses to supplementary materials as appropriate). As just one example, I suggest omitting the analyses of performance on the secondary task from the main text.

2.  I also wondered if the introduction could be shortened. The paper is currently themed around the idea of cumulative cultural evolution, but ultimately comes down to a question about cognitive processes that need not be tied to CCE – much as you note at lines 179-182. On that basis, it felt like some of the background on CCE might perhaps be superfluous (e.g., lines 95-103; lines 113-122). Also, on the topic of length: (1) the abstract should be 300 words max, and (2) you could omit Table 4, given that these data are also shown in Figure 3. Likewise the table shown at line 722 (which does not have a table number).

3.  Reviewer 2 notes that “the task simulates making inferences with social information, but is not a social learning task in itself”. I agree, and found the title a little misleading in this regard: it led me to expect that the study would be investigating (specifically) social learning strategies.

4.  I don’t think it’s appropriate to describe the 2AFC task as being a “search task”. Visual search has a specific meaning in the cognitive literature, Wikipedia gets it about right: “Visual search is a type of perceptual task… that typically involves an active scan of the visual environment for a particular object or feature (the target) among other objects or features”, e.g., searching for a red item among green items. Your task is more about choosing an option than finding an option; it would be better described as a “binary response” task or a “binary choice” task than a search task.

5.  Am I right in thinking that the WSLS condition in Experiment 2 is the only condition out of both experiments in which participants could not always deduce the correct answer(s)? As I understand it, if they are presented with one correct answer and one incorrect answer (as in the example in Fig 5), they would have to guess (with 50% success rate) one of the correct locations. If so, please could that be clarified.

6.  Line 735: “The EMCC would predict that the dual-task would have more of an impact in the WSLS group, so would therefore predict longer RTs in the EF block of the selective group, when compared to the control and to the WS group. Longer RTs would also be predicted in the EF block compared to the control block, and when under an additional working memory load. These predictions are supported by the significant interaction between group, block condition and memory load”. The last sentence here is potentially misleading, since it implicitly attributes the source of the three-way interaction to the set of predictions that has just been described, and ignores the fact that there are unpredicted aspects of the findings that are also contributing to this interaction. For example, under no-memory-load conditions, the EF dual task seems to create an impairment for the WS task but not the WSLS task. It’s hard to see how this follows from the EMCC.

7.  A few more minor / specific points:

-  Line 446: “Participants who passed the training round then completed a further 72 trials of testing in each condition in full dual-task conditions”. When I first read this I assumed that “each condition” referred to learning-strategy conditions which was to be manipulated within-subjects, but having read on I guess this must instead have referred to EF and control versions of the dual task. Is that correct, and if so please could it be clarified in the text?

-  Please could it be clarified in the text that when you are talking about the “audio task”, you are talking about the switching task that’s being used as a secondary task?

-  Line 495 (and other places): “A post-hoc Bonferroni-Holm correction of all pairwise comparisons of group membership indicated” should be something like “Post-hoc pairwise comparisons of group membership using Bonferroni-Holm correction indicated…”

-  The Figure 3 caption is too ready to accept the null hypothesis. This should be changed to something more like: “There was no SIGNIFICANT difference overall between the LS and WS groups. The difference between the control and EF block condition was NOT SIGNIFICANTLY DIFFERENT for each strategy group.”

-  We probably don’t need all the detail on what happened during the between-block breaks (lines 644-650). It’s enough to say that participants took a 5-min break between phases.

We look forward to receiving your revised manuscript.

Kind regards,

Mike Le Pelley

Academic Editor

PLOS ONE

Journal Requirements:

2.  1) Please consider changing the title so as to meet our title format requirement (https://journals.plos.org/plosone/s/submission-guidelines). In particular, the title should be "Specific, descriptive, concise, and comprehensible to readers outside the field" and in this case it is not informative and specific about your study's scope and methodology.

2) You indicated that you had ethical approval for your study. In your Methods section, please ensure you have also stated whether you obtained consent from parents or guardians of the minors included in the study or whether the research ethics committee or IRB specifically waived the need for their consent.

3) In your manuscript, please include the information removed for blind review since PLOS ONE's peer-review process is not double blinded (https://journals.plos.org/plosone/s/editorial-and-peer-review-process).

3. Our internal editors have looked over your manuscript and determined that it is within the scope of our Cognitive Developmental Psychology Call for Papers. The Collection will encompass a diverse range of research articles in developmental psychology, including early cognitive development, language development, atypical development, cognitive processing across the lifespan, among others, with an emphasis on transparent and reproducible reporting practices.  Additional information can be found on our announcement page: https://collections.plos.org/s/cognitive-psychology.  If you would like your manuscript to be considered for this collection, please let us know in your cover letter and we will ensure that your paper is treated as if you were responding to this call. Please note that being considered for the Collection does not require an additional peer review beyond the journal’s standard process and will not delay the publication of your manuscript if it is accepted by PLOS ONE. If you would prefer to remove your manuscript from collection consideration, please specify this in the cover letter.

4. Please amend the manuscript submission data (via Edit Submission) to include author: Juliet Dunstone.

5. Please amend your authorship list in your manuscript file to include author: Juliet Robertstone.

Reviewers' comments:

Reviewer's Responses to Questions

**Comments to the Author**

1. Is the manuscript technically sound, and do the data support the conclusions?

Reviewer #1: Yes

Reviewer #2: Yes

2. Has the statistical analysis been performed appropriately and rigorously? 

Reviewer #1: Yes

Reviewer #2: Yes

3. Have the authors made all data underlying the findings in their manuscript fully available?

Reviewer #1: Yes

Reviewer #2: Yes

4. Is the manuscript presented in an intelligible fashion and written in standard English?

Reviewer #1: Yes

Reviewer #2: Yes

5. Review Comments to the Author

Reviewer #1: The paper presents a series of cleverly and intricately designed and analysed studies to address the extent to which executive functions (as a proxy for explicit metacognition) are involved in flexible use of social learning strategies. The paper is very lengthy and may be better presented as two good and necessary preliminary studies (the current pilot study and study 1 which is not super representative of the cumulative culture context and had an issue with ceiling effects) with much of the text in supplementary information. This would allow focus on the more ecologically valid study 2 which also incorporated valuable elements of measuring memory load and, crucially –given the focus of the introduction- metacognition. Given this latter focus the authors are careful not to over-claim from their results and the conclusion, abstract and title reflect this. Given the aim of the paper it would be interesting to hear why a more obviously metacognitive proxy for a flexible social learning strategy underlying cumulative culture was not used, rather than the WSLS paradigm. For example, where an individual has to judge the quality of their own knowledge/social information before determining whether to adopt new social information/stick with their own knowledge (and or innovate). There are many clarity issues that are easily resolved with consistent use of terminology (see below comments).

L47 – it would be helpful to explain what distinguishes the first from the second study (what is different if anything?).

L50 – it is not until this line of the abstract that any mention of measuring metacognitive ability is made. This is odd as the abstract introduces the point of the study as being to do with testing the hypothesis about human metacognitive ability and our cultural abilities. [perhaps the flexible strategy condition equates to increased metacognition but this is not stated].

L56 – use past tense. (‘…. could have been ….’)

L57 – what does “such contexts” refer to? This is not clear.

L84 – doesn’t make sense ‘… away from unhelpful…’

L102-3 – a caveat should be added here that obviously being able to ‘verbalise’ a strategy is not the criteria that distinguishes humans from nonhumans regarding metacognition as non-human animals cannot speak. Also, there has been a fair amount of research into metacognition in nonhumans that really ought to be mentioned and incorporated into your argument. I realise that you are not attempting a comparative study or attempting to address the ‘human unique’ aspect of the theory (it would be good to state this explicitly early on) but the intro discussion of the theory seems incomplete as is.

L151 – it is a bit presumptious to assume that absolutely all nonhuman SLSs have to be non-explicit. We haven’t really tested this properly in nonhumans. Use of words like ‘likely differentiates’ or ‘may differentiate’ would be more appropriate.

Table 1: working memory task – clarify what the ‘test trial’ refers to as it is not mentioned in this way in the rest of the table (consistency in terms).

L318 & L472 – clarify why removal of outliers important for these data.

L368 – without comparing the influence of concurrent EF tasks to social and non-social WSLS tasks, can you actually say that working memory and switching are specifically involved in the use of social info.?

L450 – check for consistency of condition terms used – strategic copying is used here but elsewhere flexible, selective etc.

L452 – replace ‘showing’ with ‘observing’ so the condition is from the point of view of the participant

L454 – clarify what ‘rewarded’ and ‘unrewarded’ means – is it that they observed the location of the desired fish or undesired shark? Is it the same as the ‘successful or unsuccessful information trials’ on L507 or ‘informational trial type’ in Fig 4?

L463 – complete relative statements “slower” than what?

L491 – what is a ‘strategy group’? I’m guessing the conditions but consistency in terms will help the reader. Similarly on L497 we get ‘response type’ conditions and on L596 there is mention of ‘repeated copying condition’.

L578 – is the WSLS really the “only condition analogous to a social learning strategy”? or did you mean the only one analogous to a SLS that could support cumulative culture? WS or LS could both be described as content-dependent SLSs.

L595 – by ‘two different strategies’ do you mean ‘both WS and LS’? If so I think the latter is clearer.

L620 - Can you clarify that participants in study 2 were task naive and hadn’t taken part in the pilot or study 1.

L680-681 – the explanation of points being ‘equal to the confidence rating given’ is not clear (procedurally or reasoning wise). Suggest move the explanation given from L800 (in the results) up to the Methods here.

L686 – ‘in conjunction’ is unclear, do you mean the metacognition trials were interspersed with the search trials and for both participants had to do the audio switching task concurrently?

L948 – the discussion/conclusion could do with returning to the argument of why explicit metacognition is thought to be important to cumulative culture and what your results have to say about this (with the appropriate caveats/cautions). Why was the conscious aspect of metacognition not addressed by asking participants whether they were aware of using a strategy and what it was?

Reviewer #2: This is a thoroughly investigated and reported study on whether some social learning strategies are explicitly metacognitive, by requiring participants to complete an executive function (EF) task while simultaneously completing a learning task that requires a flexible learning strategy. the authors report a pilot study and two experiments. In experiment 1 the EF shadowing task had a comparable effect on both the flexible learning task and two control tasks. In experiment 2 the difficulty of the flexible learning task was increased by incorporating a memory component and increasing the number of stimuli to be learned about in each trial. In this case there was a clearer effect on the flexible learning task than on the control task. The results are important within the context of recent theories of human cumulative cultural evolution.

The rationale for the study is very clearly and concisely put forward in the introduction. However, although I can understand the authors' thoroughness in reporting their study, the overall report is long and rather bloated. Essentially this study asks a simple question - does an executive function task affect a learning task in which the participant has to respond flexibly more than a task in which they don't. There is a lot of methods and results material for the reader to get through to find out the answer to this, and my main recommendation is to cut some of the report to make the story easier for the reader to follow. To that end, I would recommend putting the pilot study into a supplement - only the most engaged reader will want to know how you came to choose the EF task you did.

The nature of the flexible learning task (win-stay lose-shift) is alluded to in numerous places in the introduction but is never explained in concrete terms. I struggled to understand what the participant actually did until I delved into the supplementary materials, so I think the WSLS task has to be much clearer front and centre.

minor points

line 113. could you be clearer when you say 'many focus on different aspects of cultural evolution that could be assessed differently'? What aspects, how could they be assessed differently? I assume you really want to say 'although there are multiple definitions of CCE, for our purposes Mesoudi and Thornton (2019) have provided to definitions we will use...'

line 369. Could you make it clear that the task simulates making inferences with social information, but is not a social learning task in itself.

line 397. The different conditions in this experiment are poorly explained in the introduction. Could you be clearer?

line 582. It did occur to me that because I didn't fully understand what the participant did in the wsls task, perhaps there was a greater motor cost in this task than in the control tasks. Can the authors be sure that the EF task doesn't simply produce an interference effect on motor control?

line 721. I don't think the table provides any information that figure 7 doesn't also show. I would ditch the table and put figure 7 much earlier in the results section.

line 731. Was baseline taken from R1 or R2?

6. PLOS authors have the option to publish the peer review history of their article (what does this mean?). If published, this will include your full peer review and any attached files.

Reviewer #1: **Yes: **Rachel L Kendal

Reviewer #2: No

---

## [Author Response · Author response to Decision Letter 0]

22 Dec 2020

Responses to reviewer and editor comments have been given in the uploaded file tagged as 'response to reviewers'

---

## [Editor Report · Decision Letter 1]

15 Jan 2021

PONE-D-20-26673R1

Limited evidence for executive function load impairing selective copying in a win-stay lose-shift task

PLOS ONE

Dear Dr. Dunstone,

Thank you for submitting this revised version of your manuscript to PLOS ONE. I have not sent this revision back to the original reviewers because their previous suggestions were clear; instead for the sake of a speedy process I have read over the manuscript carefully myself, with their comments in mind.

You have done a good job of responding to the issues raised regarding the previous submission, and I suspect this more streamlined version of the paper will be more accessible to readers. I have noted below a number of minor points that came up in my reading of this revision; all of them should be straightforward to address (some are tiny typos since PLOS does not have copy-editing).

1.  L211-212: some note should be made here of the dual task, which is a critical aspect of Experiment 1. E.g., perhaps something like “Experiment 1 therefore compared participant performance in a basic selection task when a selective strategy was required, compared to when it was not, under conditions in which participants performed a secondary task that was intended to place a high or low load on executive resources”.

2.  L220: WS and LS should probably be defined in their ‘separate’ form (WSLS has previously been defined, but a reader may not make the connection without prompting).

3.  L263: It’s not clear to me what “vicariously presented information” means here. The information is presented directly to participants; they are not told information via a third party. I appreciate that an analogy is being made here to social learning, where a participant might see the outcome of somebody else’s behaviour. But regardless, in the current task the relevant outcome information is presented directly to participants.

4.  A general point regarding terminology relating to the audio task. In some places the conditions of this task are referred to as the “switching task” and “control task”, whereas in other places they are referred to as the “executive function / EF task” and “control task” (and notably, when the task is introduced in Methods [L276-288] there’s no mention that the switching task is also going to be called the EF task). Please can you make the terminology consistent throughout the paper, which will help the reader to keep track of which tasks you are talking about. On this topic, “switching task” seems like perhaps a better label than “EF task”, given that many of the tasks that you talk about in this study would place a load on executive function.

(Also on this point, in the abstract this talk is referred to as a “distractor task”, which is different yet again.)

One more (related) terminological issue that is somewhat pedantic. “Dual” means “consisting of two things”, so it doesn’t really make sense to call the audio task the “dual task”, because it is only a single thing. Instead it should probably be termed a “secondary” task. That is, the whole procedure is a dual-task design as it consists of two separate tasks, a primary task (e.g., WSLS) and a secondary task (e.g., audio switching).

5.  Reaction time is sometimes written out in full, and sometimes as “RT”. Given that your article already has a lot of abbreviations, I recommend writing it out in full each time.

6.  L374-375: what was the correction for multiple comparisons that was used here? Bonferroni-Holm (like in the case mentioned at line 378)?

7.  L383, “the *switching *task had a negative impact on response times in each group”: please state what the negative impact is relative to (presumably the control task).

8.  L408: This section is titled “Executive Function Dual Task” but talks about data from both the switching (EF) task and the control task (which is being considered as a non-EF task).

9.  L412: Please write out “Supplementary Information” in full.

10.  L432-433: Please change to  “accuracy in the concurrent audio *switching *task also decreased (see Supplementary Information)”, since that is where the analysis of switching-task RTs is reported.

11.  L580-589: To avoid unnecessary repetition, this section could be replaced with “Overall 531 outliers were removed from the data for very long or very short reaction times, using the same procedure for outlier removal as in Experiment 1”.

12.  L615-617: “The EMCC would predict that the dual-task would have more of an impact in the WSLS group, so would therefore predict longer RTs in the EF block of the selective group, when compared to the control and to the WS group”. The impact of the dual task is defined by the impairment in the switching (EF) condition relative to the control (non-EF) condition. So should the second part of this sentence instead be something like: “so would therefore predict a greater performance impairment in the EF block (relative to the control block) for the selective (WSLS) group than for the WS group”?

13.  L759:  “There was a significant decrease in concurrent audio *switching *task accuracy for both main task conditions, especially the metacognition task” – a decrease relative to what? Also, should there be a reference to Supplementary Information here?

14.  L779: “tasking switching” is a typo.

15.  L257 and L496: “Psychopy” should be written “PsychoPy”.

A rebuttal letter that responds to each point raised by the academic editor. You should upload this letter as a separate file labeled 'Response to Reviewers'.A marked-up copy of your manuscript that highlights changes made to the original version. You should upload this as a separate file labeled 'Revised Manuscript with Track Changes'.An unmarked version of your revised paper without tracked changes. You should upload this as a separate file labeled 'Manuscript'.

We look forward to receiving your revised manuscript.

Kind regards,

Mike Le Pelley

Academic Editor

PLOS ONE

---

## [Author Response · Author response to Decision Letter 1]

1 Feb 2021

My response to editor is included in the 'response to reviews' rebuttal letter attached in the submission documents.

---

## [Editor Report · Decision Letter 2]

3 Feb 2021

Limited evidence for executive function load impairing selective copying in a win-stay lose-shift task

PONE-D-20-26673R2

Dear Dr. Dunstone,

We’re pleased to inform you that your manuscript has been judged scientifically suitable for publication and will be formally accepted for publication once it meets all outstanding technical requirements.

Kind regards,

Mike E. Le Pelley, Ph.D.

Academic Editor

PLOS ONE
---

## [Editor Report · Acceptance letter]

17 Feb 2021

PONE-D-20-26673R2 

Limited evidence for executive function load impairing selective copying in a win-stay lose-shift task 

Dear Dr. Dunstone:

I'm pleased to inform you that your manuscript has been deemed suitable for publication in PLOS ONE. Congratulations! Your manuscript is now with our production department. 

Kind regards, 

on behalf of

Dr. Mike E. Le Pelley 

Academic Editor

PLOS ONE